# Application of Plant Oils as Functional Additives in Edible Films and Coatings for Food Packaging: A Review

**DOI:** 10.3390/foods13070997

**Published:** 2024-03-25

**Authors:** Hooman Chodar Moghadas, Ruchi Chauhan, J. Scott Smith

**Affiliations:** Food Science Institute, Kansas State University, Manhattan, KS 66506, USA; hoomanmoghadas@ksu.edu (H.C.M.); ruchichauhan@ksu.edu (R.C.)

**Keywords:** edible films, edible coatings, biopolymers, plant oils, biodegradable, food preservation

## Abstract

Increasing environmental concerns over using petroleum-based packaging materials in the food industry have encouraged researchers to produce edible food packaging materials from renewable sources. Biopolymer-based edible films and coatings can be implemented as bio-based packaging materials for prolonging the shelf life of food products. However, poor mechanical characteristics and high permeability for water vapor limit their practical applications. In this regard, plant oils (POs) as natural additives have a high potential to overcome certain shortcomings related to the functionality of edible packaging materials. In this paper, a summary of the effects of Pos as natural additives on different properties of edible films and coatings is presented. Moreover, the application of edible films and coatings containing POs for the preservation of different food products is also discussed. It has been found that incorporation of POs could result in improvements in packaging’s barrier, antioxidant, and antimicrobial properties. Furthermore, the incorporation of POs could significantly improve the performance of edible packaging materials in preserving the quality attributes of various food products. Overall, the current review highlights the potential of POs as natural additives for application in edible food packaging materials.

## 1. Introduction

One of the significant challenges facing the food industry today is reducing the increasing production of food waste. Globally, one-third of the total produced food, equivalent to about 1.3 billion tons, goes to waste annually [1]. It is estimated that, by 2025, the generated waste will exceed 2.2 billion tons each year [2]. Therefore, food waste reduction is an issue of great importance as it can prevent economic losses and environment pollution.

Food packaging provides a novel approach to minimize food wastage through protecting food products from physical, chemical, and microbial deterioration. Currently, polymers derived from petroleum, such as polyethylene, polypropylene, and polyvinyl chloride, are commonly used in food packaging [3]. However, since these materials are non-recyclable, their use can give rise to serious environmental and health-related problems [4]. As a result, replacing synthetic materials with environmentally friendly alternatives is crucial. In recent years, numerous studies have demonstrated that edible packaging materials have a high potential to replace petroleum-based polymers in the food industry [5]. The major advantage of using edible packaging in the food industry is that they are non-toxic, recyclable, and eco-friendly. Edible packaging not only helps to maintain the quality of food products but can also have a positive impact on the visual appearance and consumer acceptability of the products [6]. Furthermore, edible packaging can be incorporated with numerous active agents such as antioxidants and antimicrobials, which can improve quality and safety of food products [5].

The application of biopolymers as base materials in the production of edible films and coatings has become more common recently, mainly because of their low price and biocompatibility [6]. Biopolymers are polymeric materials that can be derived from plants, animals, or microbial sources. So far, various biopolymers have been studied with the aim of producing edible films or coatings. Among them, polysaccharides and proteins are the most common biopolymers used for this purpose. Chitosan, gums, cellulose, and starch are among the most popular polysaccharides used in edible packaging, whereas the most commonly used proteins in edible packaging are gelatin, whey protein, egg albumin, casein, and soy protein [7]. Despite their good film-forming ability, they have high water vapor permeability owing to their hydrophilic nature. Different strategies could be applied to decrease the water vapor permeability of biopolymer-based packaging materials; among them, incorporation of hydrophobic components into the biopolymer matrix is the most common. In this context, hydrophobic components such as essential oils and plant oils (POs) have been reported to decrease the water vapor permeability of biopolymer-based packaging systems [8]. POs, obtained from different types of oil-containing seeds, fruits, and nuts, are lipid extracts rich in fatty acids. The most commonly used techniques for extraction of them are solvent or mechanical extraction methods. On the other hand, essential oils are highly concentrated extracts derived from specific plant parts like leaves, flowers, or stems using processes like steam distillation and hydrodistillation. Essential oils are rich in aromatic compounds that give them distinctive aromas and scents. Since essential oils generally have an intense aroma, their presence in edible films and coatings might impose undesirable changes in the sensory quality of food products [9].

POs are an essential part of the human diet due to their significant role in improving human health. Apart from being a primary source of essential fatty acids (linoleic and linolenic), POs have numerous bioactive compounds in their composition that are responsible for their antioxidant and antimicrobial activities [10]. While bioactive compounds are typically more abundant in essential oils, certain plant oils also contain noteworthy amounts of these compounds. For instance, tocopherols, carotenoids, sterols, and phenolic compounds found in various plant oils such as olive [11], pomegranate seed [12], and camellia [13] oils play an important role in conferring significant biological properties to them. In the last decade, POs from various sources have been extensively explored as functional additives in biopolymer-based packaging materials. Owing to their hydrophobic character, they are expected to improve the water vapor barrier properties of these packaging systems. In addition, the incorporation of POs into these packaging materials has been shown to positively influence the mechanical, antimicrobial, and antioxidant properties of the packaging system [14,15,16]. Nevertheless, there is no review on the application of POs in biopolymer-based food packaging materials. Therefore, the present paper reviews the application of POs as functional additives in biopolymer-based packaging materials, including their impact on the mechanical, physical, barrier, antimicrobial, and antioxidant properties of them. Moreover, the influence of PO addition to biopolymer-based packaging materials on the quality attributes of various food products is also reviewed. This paper provides valuable information for the food packaging industry regarding the application of POs as functional additives in edible food packaging.

## 2. Biopolymer-Based Films and Coatings Incorporated with Plant Oils

Edible films and coatings are gaining increasing attention in the food packaging industry due to their sustainability and eco-friendliness. The components employed to make edible films and coatings should essentially be food-grade, including not only the polymer matrix but also the solvent and any other potential additives. Despite the fact that edible films and coatings serve the same function, the way that they are produced and applied is quite different (Figure 1). Edible films are used for wrapping the food product, while edible coatings are applied to food products in liquid form using dipping or spraying methods [17]. Another main difference between them is that the thickness of edible coatings is generally lower (less than 0.3 mm) than that of the edible films [5]. For effective food packaging, edible films require minimum mechanical properties such as tensile strength (TS) to resist breaking, elongation at break (EB) for stretchability, and flexibility for bending without compromising integrity. On the other hand, edible coatings must exhibit resistance to environmental conditions such as humidity, temperature variations, air oxidation, and light exposure. For instance, stability against air oxidation is crucial to prevent the deterioration of coating materials, ensuring their effectiveness in preserving food quality. Additionally, resistance to light exposure helps prevent the degradation of the coating, which could compromise its protective properties through the photo-oxidation of bioactive compounds present in the coating material. Achieving a well-balanced combination of the mentioned requirements is crucial for the successful development and application of edible films and coatings for food packaging. For producing edible films and coatings, at least three components are required: solvent (usually water), a film-forming biopolymer, and plasticizer. In addition, to enhance the functionality of the package, certain additives are incorporated into the biopolymer matrix. Antioxidant and antimicrobial compounds are the most common additives used in the formulation of edible films and coatings to promote food preservation.

POs are generally used in cooking, baking, and food formulations. However, they have also recently found applications as natural additives in the production of edible films and coatings. POs have the potential to exhibit antimicrobial activity against various foodborne pathogens such as *Staphylococcus aureus*, *Salmonella enterica*, *Escherichia coli*, and *Listeria monocytogenes* [18]. In addition, POs can contain numerous biologically active compounds such as polyphenols, tocopherols, and flavonoids that can impart antioxidant and antimicrobial properties to the packaging material [10,19]. However, the beneficial effects of POs can vary greatly depending on their chemical composition. The most common POs used in the formulation of edible packaging materials are olive, sunflower, corn, and canola oils. POs can be added to biopolymer-based films and coatings either via emulsification technology or by laminating the films with an oil layer [20]. In the emulsification technique, POs are dispersed and homogenized within biopolymer aqueous solution containing suitable surfactants for forming stable emulsions. Alternatively, another emulsification method involves initially preparing an emulsion of POs with an appropriate surfactant followed by the subsequent blending of this emulsion with the biopolymer aqueous solution. The resulting biopolymer/PO emulsion from both techniques is then used either as a coating solution or for the production of an edible film (Figure 1).

## 3. Plant Oils and Biopolymers Interactions

The interactions between POs and biopolymers in edible films and coatings can play an important role in determining the overall properties and performance of the packaging material. These interactions involve molecular compatibility between the hydrophobic nature of POs and the hydrophilic characteristics of biopolymers. In this regard, the incorporation of suitable surfactants can facilitate the dispersion and homogenization of POs within the biopolymer matrix. The type and amount of POs incorporated into edible films and coatings can also significantly influence their interactions with biopolymers, consequently affecting the overall characteristics of the developed packaging material. Due to variations in fatty acid profiles, different POs exhibit varying degrees of hydrophobicity, which can influence their ability to disperse uniformly within hydrophilic biopolymer matrices. The amount of POs added to the formulation further modulates the interactions by impacting factors like flexibility, permeability, and stability of the packaging material. Excessive amounts may lead to phase separation or compromise the structural integrity of the packaging material, while an optimal amount can enhance homogeneity and maximize the synergistic effects between POs and biopolymers. The type of biopolymer is also another important factor that can influence their interactions with POs. Different biopolymers have distinct structural and chemical characteristics that can impact their compatibility with various types of POs. The hydrophilic or hydrophobic nature of the biopolymer can affect the dispersion and emulsification of POs within the matrix. Additionally, the presence of functional groups on the biopolymer chains may facilitate or hinder interactions with specific components of POs. Overall, a good understanding of these interactions is necessary for the development of effective edible films and coatings incorporating POs.

## 4. Effect of Plant Oil Incorporation on Various Properties of Edible Packaging

### 4.1. Thickness

Film thickness is an important parameter that can influence other properties of the packaging system such as water vapor permeability (WVP) and mechanical characteristics [21]. Generally, the addition of POs increases the thickness of biopolymer-based packaging materials (Table 1). As an example, Hanani [22] observed an increase in the thickness of carrageenan film after the addition of olive, corn, soybean, and sunflower oils. Similarly, Niknam et al. [23] reported an increase in the thickness of *Plantago major* seed gum film after the addition of different POs including olive, maize, and canola. The increase in the thickness of the films might be due to the conformational changes in the film structure induced by the incorporation of POs [22]. Nevertheless, it should also be noted that the increase in the film thickness is not always dependent on the concentration of the applied POs. For instance, Hasan et al. [24] found that the thickness of the film based on starch/chitosan was significantly increased after the addition of olive oil at 2% (*w*/*w*). However, a further increase in the concentration of olive oil (up to 5%, *w*/*w*) did not significantly influence the film thickness. On the other hand, some studies have also shown that the addition of POs does not significantly influence the thickness of the film. For instance, Fangfang et al. [25] observed this phenomenon in starch film after the addition of coconut oil. These results show that the effect of PO addition on the thickness of film can vary greatly depending on the composition of the film matrix. One important factor to note is that, while film thickness can be influenced by its composition, this parameter can also be affected by the handling process and individuals involved in film production.

### 4.2. Mechanical Properties

Good mechanical properties are essential for edible films, since poor mechanical strength and flexibility can result in film network breakage during storage or shipment of packaged food. The mechanical properties of edible films are determined mainly by their EB and TS. EB shows the stretchability and flexibility of the film, while TS shows the strength of the film against stress. EB and TS are highly influenced by the chemical structure and composition of the films [26]. The incorporation of POs into edible films significantly affected their EB and TS (Table 1). Studies have shown that the addition of POs in edible films has diverse effects on their tensile properties. As an example, adding rapeseed oil improved TS of whey protein-based edible film [27]. Similarly, adding olive oil improved the TS of the film made from chitosan [28]. This effect could be due to the cross-linking reactions between the biopolymer and the applied oil, which decreases the free volume and molecular mobility of the film [27,28]. However, the incorporation of rapeseed oil decreased the TS of a soy-protein-isolate-based film [29]. Olive oil has also been found to decrease the TS of potato starch film [30]. Therefore, the changes in the TS of biopolymers induced via the addition of POs are strongly dependent on the type and addition level of both the biopolymer and PO. On the other hand, the addition of POs has been shown to increase the EB of biopolymer-based films (Table 1). For instance, olive oil addition increased the EB of edible films based on starch/chitosan [24], chitosan [28], gelatin [31], and albumin [32]. Niknam et al. [23] reported that the EB of the film based on *Plantago major* seed gum was significantly increased by the incorporation of various types of POs including canola, maize, and olive. These observations could be ascribed to the plasticizing effect of POs on biopolymer-based films [23]. For edible films, a high EB is preferred as it enhances the film’s ability to wrap the food product. Overall, the structural characteristics of biopolymers, the compositional variability of different POs, and the proportions of each of the components in the packaging matrix all have a significant influence on the film’s mechanical properties.

### 4.3. Solubility in Water

The water solubility of packaging materials indicates their hydrophobicity, which is a critical factor when it comes to their application for the preservation of high-moisture food products. Previous findings showed that the incorporation of POs in edible films decreased their water solubility (Table 1). As an example, Xiao et al. [16] observed that the water solubility of konjac glucomannan/agar/gum Arabic film was significantly decreased with an increase in the concentration of coconut oil, which varied from 45.2% (control) to 23.1% with the addition of 0.6% (*w*/*w*) of oil. A similar effect was observed in another study in which false flax seed oil was added to chitosan film at a concentration of 2% (*v*/*v*), and the authors observed a significant decrease in the water solubility of the film from 55% to 35% [33]. Lee et al. [34] also reported that the incorporation of 2% (*w*/*w*) sunflower seed oil to the film based on starch resulted in a reduction of 34.8% in the water solubility of the film. These results seem obvious, given that the inclusion of a hydrophobic compound in the biopolymer matrix can reduce potential interactions among water molecules and the polymer matrix. Therefore, POs can be considered as potential additives to enhance the hydrophobicity of biopolymer-based films and coatings.

**Table 1 foods-13-00997-t001:** The influence of plant oil addition on different properties of edible films.

Matrix	Plant Oils	Matrix: Oil Proportion	Findings	Refs
Alginate	Linseed oil	Film coated with oil (thickness of the oil layer: 0.5 µm)	Decreased WVP; increased TS, UV barrier ability, thermal stability, and thickness	[3]
	Sunflower seed oil	Film coated with oil (thickness of the oil layer: 0.5 µm)	Decreased WVP; increased UV barrier ability, total color value, thermal stability, and thickness	
Alginate	Soybean oil	1% (*w*/*v*): 0.5–1.5% (*v*/*v*)	Decreased WVP (over the range of 1% to 1.5% of soybean oil), TS, and EB; increased opacity	[35]
Carboxymethyl cellulose	Black cumin seed oil	2% (*w*/*v*): 0–1% (*w*/*v*)	Decreased water solubility and EB; increased TS, antioxidant activity, thermal stability, and total color difference	[36]
Chitosan (medium MW)	*Berberis crataegina* seed oil	1% (*w*/*v*): 0–1.6% (*v*/*v*)	Decreased water solubility, TS, and thermal stability; increased EB, opacity, antioxidant activity, UV barrier ability, and thickness	[37]
Chitosan (MW: 161 kDa)	Olive oil	2% (*w*/*v*): 0–15% (*w*/*w*, based on the mass of chitosan)	Decreased WVP; increased TS, EB, thickness, and opacity	[28]
Chitosan (medium MW)	False flax seed oil	2% (*w*/*v*): 0–2% (*v*/*v*)	Decreased water solubility; increased EB, opacity, thermal stability, and antioxidant activity	[33]
Corn starch	Sunflower oil	3% (*w*/*v*): 0–10% (*w*/*w*, based on the mass of starch)	Decreased water solubility, WVP, and TS; increased opacity	[38]
Corn starch	Coffee oil	4% (*w*/*v*): 0–1% (*v*/*v*)	Decreased WVP and opacity; increased TS, EB, UV barrier ability, and thickness	[14]
Corn starch/sodium alginate/gum Arabic	Coconut oil	2% (*w*/*w*): 0–70% (*w*/*w*, based on the mass of gum Arabic)	Decreased OP, WVP, and TS; increased EB and thickness	[39]
Fenugreek galactomannan/xanthan gum	Grape seed oil	1.5% (*w*/*v*):0–0.5% (*v*/*v*)	Decreased OP, WVP, TS, opacity, and thermal stability; increased EB and thickness	[21]
Konjak glucomanan/agar/gum Arabic	Coconut oil	2.4% (*w*/*w*): 0–0.6% (*w*/*w*)	Decreased water solubility, WVP, and TS; increased EB and thickness	[16]
Mung bean starch/guar gum	Sunflower seed oil	2.75% (*w*/*w*): 0–2% (*w*/*w*)	Decreased water solubility, WVP, TS, and EB; increased opacity, OP, and total color difference	[34]
*Plantago major* seed gum	Canola oil	1% (*w*/*v*): 0–0.5% (*v*/*v*)	Decreased WVP, TS, thermal stability; increased EB and thickness	[23]
	Maize oil	1% (*w*/*v*): 0–0.5% (*v*/*v*)	Decreased WVP, TS, and thermal stability; increased EB and thickness	
	Olive oil	1% (*w*/*v*): 0–0.5% (*v*/*v*)	Decreased WVP, TS, and thermal stability; increased EB and thickness	
Potato starch	Coconut oil	2.5% (*w*/*w*): 0–112% (*w*/*w*, based on the mass of potato starch)	Decreased WVP and thermal stability; increased TS (up to 14% of coconut oil) and transparency	[25]
Potato starch	Olive oil	5% (*w*/*v*): 0–10% (*w*/*w*, based on the mass of potato starch)	Decreased WVP, TS, EB, and thermal stability; increased total color difference and UV barrier ability	[30]
Gelatin	Corn oil	4% (*w*/*v*): 0–1.2% (*w*/*v*)	Decreased water solubility, WVP, TS, and transparency; increased EB, thermal stability, and UV barrier ability	[40]
Albumin	Olive oil	9% (*w*/*v*): 0–1.5% (*v*/*v*)	Decreased water solubility, WVP, and OP; increased TS, EB, opacity, total color difference, and thickness	[32]
Gelatin	Camellia oil	3% (*w*/*v*): 0–100% (*w*/*w*, based on the mass of gelatin)	Decreased WVP, TS, and thermal stability; increased EB, UV barrier ability, antioxidant activity, total color difference, opacity, and thickness	[15]
Gelatin	Olive oil	5% (*w*/*w*):0–20% (*w*/*w*, based on the mass of gelatin)	Decreased WVP; increased TS, EB, thermal stability, opacity, and UV barrier ability	[31]
Gelatin	Sunflower oil	4% (*w*/*v*):0–1% (*w*/*v*)	Decreased water solubility, WVP, and transparency	[41]
Soy protein isolate	Rapeseed oil	10% (*w*/*w*): 0–3% (*w*/*w*)	Decreased WVP and TS; increased EB, total color difference, and opacity	[29]
Soy protein isolate	Flaxseed oil	5% (*w*/*w*):1–10% (*w*/*w*)	Decreased WVP; increased TS (up to 5% of flaxseed oil), EB, and total color difference	[42]
Whey protein isolate	Sunflower oil	8% (*w*/*w*): 0–0.15% (*w*/*w*)	Decreased water solubility, WVP, OP (up to 0.05% of sunflower oil), and TS; increased EB, thermal stability, and opacity	[43]
Whey protein isolate	Almond oil	8% (*w*/*w*): 0–1% (*w*/*w*)	Decreased WVP; increased solubility in water, TS (up to 0.5% of almond oil), EB (at 1% of almond oil), total color difference, opacity, and OP	[44]
	Walnut oil	8% (*w*/*w*): 0–1% (*w*/*w*)	Decreased water solubility, WVP, and EB; increased TS (up to 0.5% of walnut oil), total color difference, OP, and opacity	
Whey protein isolate	Rapeseed oil	8% (*w*/*w*): 0–3% (*v*/*v*)	Decreased water solubility; increased TS, EB, total color difference, and opacity	[27]

WVP: water vapor permeability, TS: tensile strength, EB: elongation at break, UV: ultraviolet, OP: oxygen permeability, MW: molecular weight.

### 4.4. Water Vapor Permeability (WVP)

The WVP of edible packaging materials plays a significant role in determining the quality and safety of food products during storage [45]. One of the key functions of edible packaging materials is to minimize water vapor transmission between the surrounding environment and food. Therefore, it is vital to investigate the WVP of them before their application for food packaging.

Generally, biopolymer-based packaging materials have limited resistance against water vapor due to their hydrophilic character [29,39]. Therefore, the incorporation of hydrophobic compounds into the polymer matrix is needed to decrease the transmission of water vapor through the film. The effect of PO addition on the WVP of edible films has been examined in several studies, and different findings have been observed (Table 1). For instance, the WVP of whey-protein-isolate-based film incorporated with sunflower oil was found to be significantly lower than that of the film without oil [43]. A decrease in WVP as a result of the addition of POs was also observed in other types of biopolymers such as chitosan [28], soy protein isolate [29], starch [38], and gelatin [15] films. POs (as hydrophobic components) can reduce the WVP of films by increasing the hydrophobicity of the film and creating tortuosity in the pathway that water molecules take to penetrate the film [20]. Nevertheless, the type and concentration of the PO are also important factors that can influence the WVP of films. As an example, Niknam et al. [23] found that maize oil was more effective than canola and olive oils in decreasing the WVP of a film based on *Plantago major* seed gum. In another study, it was found that walnut oil has a greater effect than almond oil on decreasing the WVP of whey-protein-isolate-based films [44]. Hopkins et al. [42] reported that the WVP of a soy-protein-isolate-based film increased as the concentration of flaxseed oil in the film formulation increased from 1% to 3% (*w*/*w*). However, further increases in the concentration of the oil (up to 10%, *w*/*w*) significantly decreased the WVP of the film. Considering these findings, the selection of optimal PO type and concentration is necessary to formulate edible packaging materials with the desired WVP.

### 4.5. Oxygen Permeability (OP)

Oxidation is responsible for quality loss in foods by destroying macro/micro-nutrients and producing off-flavors [46]. Therefore, the development of packaging materials with ideal OP is important in protecting food products against deterioration.

The addition of POs either decreased or increased the OP of the films (Table 1). As an example, the addition of sunflower seed oil increased the OP of a film based on mung bean starch/guar gum [34]. Similarly, films of whey protein isolate showed a significant increase in OP when fortified with walnut or almond oils [44]. These observations might be explained by the fact that oxygen is a non-polar molecule and, thus, its solubility in films containing oil is higher than in films without oil addition [34]. Nevertheless, a significant decrease in the OP of whey protein isolate film was observed after the addition of sunflower seed oil [43]. Similarly, Xiao et al. [39] reported that the addition of coconut oil decreased the OP of the film based on corn starch/sodium alginate/gum Arabic. Xiao et al. [39] explained that the addition of coconut oil increased film thickness and, thus, the OP of the developed film decreased. Overall, it can be concluded that the effect of PO addition on the OP of films highly depends on the composition and type of both the PO and biopolymer.

### 4.6. Ultraviolet Light Transmittance

The ultraviolet (UV) barrier property of packaging materials can play a significant role in protecting foods against the deterioration of their ingredients. UV light can accelerate the degradation of food ingredients via photochemical reactions [14]. Thus, the development of food packaging materials with a high UV-blocking capacity can help to minimize the quality loss of packaged foods.

The UV barrier ability of biopolymer-based packaging materials depends, to a great extent, on the type of the polymer as well as the presence of UV-blocking materials (such as nanomaterials and polyphenols) in the film formulation [47]. According to the published studies, the addition of POs improves the UV barrier ability of biopolymer-based films and coatings (Table 1). Jusoh et al. [48] reported that gelatin film incorporated with coconut oil had a significantly higher UV barrier ability than pure gelatin film. The authors attributed this result to the UV-absorbing ability of the antioxidant compounds present in coconut oil. A similar result was observed for gelatin films with the addition of olive oil [31], camellia oil [15], and corn oil [40]. Farajpour et al. [30] observed that the starch film containing olive oil had a higher UV-light barrier property than the starch film alone. Nehchiri et al. [3] found that the UV-light barrier ability of the alginate film was significantly improved when the film was coated with sunflower or linseed oil. POs typically contain various polyphenolic compounds that can act as UV light blockers and improve the UV protection properties of the film [47,49]. Therefore, they can be considered as potential UV barrier ingredients for application in food packaging.

### 4.7. Optical Properties

The optical properties of food packaging materials can have a significant influence on the visual appearance of the product as well as purchase intention of consumers. These properties can also play an important role in protecting food products against light-induced deterioration [50]. Thus, evaluation of these properties is of high importance in the development of food packaging materials. Color and transparency are among the most important optical properties of biopolymer-based packaging materials. The color of the films is generally expressed using color parameters (*L**, *a**, and *b**) and total color difference, which is a measurement of the change in color as calculated based on the values of color parameters [34]. The transparency value, which is often referred to as “opacity”, is also calculated by dividing the absorbance of the film at 600 nm by its thickness [15].

The incorporation of POs typically makes edible films and coatings colored and less transparent (Table 1). As an example, the incorporation of walnut and almond oils increased the opacity and total color difference of the film based on whey protein isolate [44]. Increased total color difference and opacity due to incorporation of POs was also observed in other studies on soy protein isolate [29], albumin [32], and gelatin [15] films. The increase in the total color difference of the films as a result of oil addition can be attributed to various types of pigments (such as chlorophylls) present in POs [23]. Additionally, the increase in the opacity of the films resulting from the addition of POs could be explained by the light scattering effect of the dispersed oil droplets in the biopolymer network [28]. However, some studies reported a decreased value of opacity in edible films in response to the addition of POs. As an example, Wang et al. [14] found that the addition of coffee oil decreased the opacity of an corn starch-based film. Similarly, Niknam et al. [21] observed that the opacity of a film based on fenugreek galactomannan/xanthan gum slightly decreased after the addition of grape seed oil. Therefore, it can be concluded that the opacity value of films depends, to a great extent, on the chemical composition and addition level of POs as well as the type of biopolymer [29].

### 4.8. Thermal Properties

The thermal properties of packaging materials demonstrate their ability to tolerate heating conditions. Thermogravimetric analysis and differential scanning calorimetry are frequently employed to assess the thermal stability of biopolymer-based packaging materials [51]. The incorporation of POs either increased or decreased the thermal stability of edible packaging materials, depending on the type of oil and biopolymer (Table 1). For example, the incorporation of grape seed oil within a fenugreek galactomannan/xanthan gum matrix decreased the melting temperature and glass transition temperature of the film [21]. The same result was observed by Niknam et al. [23] for a film based on *Plantago major* seed gum incorporated with different POs including canola, maize, and olive. Atta et al. [52] found that the addition of olive oil to carboxymethylcellulose/bacterial cellulose film adversely affected its thermal stability as the weight loss of the film with olive oil was observed to be higher than that of the control film in thermogravimetric analysis. These results might be due to the fact that the addition of POs (as hydrophobic substances) elevate the free volume within the film structure and, thus, result in the disruption of intermolecular hydrogen bonds among polymer chains [30]. Nevertheless, some studies found that the inclusion of POs enhanced the thermal stability of biopolymer-based packaging materials (Table 1). In a study by Biswas et al. [36], it was observed that the thermal stability of carboxymethyl cellulose film reinforced with ZnO nanoparticles was significantly improved after the addition of black cumin seed oil. They attributed this finding to the increased intermolecular interactions among the film components. Several other studies have also reported an improvement in thermal stability following the addition of POs to different biopolymers such as gelatin [31,40] and chitosan [33] films.

### 4.9. Antioxidant Properties

Oxidation reactions in food products can give rise to the formation of compounds that degrade product quality and negatively affect the sensory properties of the product. Biopolymer-based packaging materials with antioxidant capacity can help to retard the oxidation of food products.

The unsaponifiable fraction of POs may contain a variety of biologically active compounds that can exert antioxidant activity [19]. Recent research has shown that the incorporation of POs into edible packaging imparted antioxidant activity [15,33,36,37]. As an example, the antioxidant activity of the chitosan film increased from 8.54% to 30.99% upon the addition of 1.6% (*v*/*v*) of *Berberis crataegina* seed oil into the film [37]. Similarly, Gursoy et al. [33] observed that the antioxidant activity of chitosan film increased from 16.09% to 23.74% as the concentration of false flax seed oil in the film increased from 0% to 2% (*v*/*v*). The antioxidant capacity of POs can vary greatly depending on their chemical composition. However, the observed antioxidant activity of them is predominantly attributed to the presence of phenolic compounds, tocols, and sterols in their chemical composition [53]. These bioactive compounds can act through different mechanisms to neutralize free radicals, inhibit oxidation reactions, and preserve the quality and sensory attributes of the food products when incorporated into edible films and coatings.

### 4.10. Antimicrobial Properties

Antimicrobial packaging can help in improving the safety of packaged foods by limiting microbial growth. In this sense, the incorporation of additives exhibiting antimicrobial activities into edible packaging materials is a promising approach for obtaining food products free from synthetic preservatives.

Recent research has shown that the incorporation of POs improved the antimicrobial ability of edible films and coatings against diverse microorganisms (Table 2). Different edible films containing different POs reduced the growth of these microorganisms to different extents, depending on the type and composition of biopolymers and POs. As an example, Akyuz et al. [54] evaluated the antimicrobial activity of the chitosan–olive/corn/sunflower oil blend films against different food-borne and human pathogen bacteria. The obtained results indicated that the chitosan film blended with olive oil or sunflower oil exhibited higher antimicrobial activity than the chitosan film blended with corn oil. POs contain numerous biologically active compounds that can act as antimicrobial agents by penetrating into a microbial cell, damaging the cell’s structural integrity, and triggering the leakage of cellular contents [5,10]. The antimicrobial activity of POs is linked to the diverse range of bioactive compounds, such as polyphenols, carotenoids, and tocopherols, in their chemical composition [55]. For instance, Medina et al. [18] found a strong correlation between the phenolic compounds in olive oil and its antimicrobial activity. Xuan et al. [56], who studied the antimicrobial activity of various POs, highlighted the variations in total phenolic and total flavonoids contents along with the individual compounds within the analyzed oils as responsible for the varied antimicrobial activity. However, it is important to note that fatty acid composition can also affect the antimicrobial properties of POs [57]. Yoon et al. [58] highlighted, in their review report, the broad spectrum of antibacterial activity of different fatty acids and monoglycerides.

In most studies, the increase in the concentration of POs in the film formulations resulted in an increase in the inhibitory activity of the films [15,25,33]. However, it is important to note that high concentrations of POs in film formulations can weaken their mechanical properties, an issue that can limit the application of films as packaging materials. Therefore, the selection of the PO and biopolymer with suitable formulation is important for obtaining a packaging material with desired mechanical and antimicrobial properties.

Most of the studies have evaluated the impact of the PO addition on the antibacterial activity of the films. However, studies on the impact of PO addition on the antifungal activity of the films are limited. In this context, Sahraee et al. [40] evaluated the impact of corn oil addition on the antifungal activity of nanochitin-incorporated gelatin film against *Aspergillus niger*. The results revealed that the antifungal activity of the film was inhibited after the addition of corn oil. The authors explained that the oil in the film matrix covers the active groups of chitin nanoparticles, thereby preventing the nanoparticles from showing antifungal activity. However, further studies are required to explore the potential antifungal activities of biopolymer-based packaging materials containing POs.

## 5. Food Applications

POs can be used as natural and safe additives to improve the performance of edible films and coatings in prolonging the shelf life of food products. Different edible films and coatings incorporated with POs were used to preserve different food products (Table 3). The incorporation of POs into edible packaging materials assisted in preserving the safety and quality of them. These results can readily be attributed to the improved antimicrobial, antioxidant, and barrier properties of the edible packaging materials as a result of PO addition.

### 5.1. Fruits and Vegetables

Fruits and vegetables have a limited shelf life due to their highly perishable nature. Therefore, it is of great importance for the food industry to improve the functionality of biopolymer-based films and coatings for the preservation of these foods. In this context, Xiao et al. [16] investigated the impact of the film based on konjak glucomannan/agar/gum Arabic containing coconut oil on the quality attributes of cucumbers stored under refrigerated conditions for 12 days (Figure 2). The results revealed that the weight loss and firmness reduction of the samples packed in the film containing coconut oil were significantly lower than those of the unpacked samples and samples packed in the film without coconut oil. Melikoğlu et al. [65] reported the application of a carboxymethyl cellulose coating incorporated with pomegranate seed oil on strawberries and observed that the addition of pomegranate seed oil to the coating helped in decreasing weight loss and retaining phenolic compounds of strawberries during storage for 16 days under refrigerated conditions. Tran et al. [59] studied the impact of a chitosan coating with tea seed oil on the qualitative characteristics of pear fruit throughout storage for 21 days at 25 °C. Samples coated with chitosan containing tea seed oil showed a lower respiration rate, higher total soluble solids, and increased resistance against fungal decay when compared to the uncoated samples and the samples coated with chitosan alone. In another study, pear fruit was coated with a soy protein isolate coating containing olive oil [66]. The authors in that study observed that the increase in the concentration of olive oil in the coating led to a decrease in the weight loss of the pear fruit after storage at ambient temperature for 5 days. Hassani et al. [61] evaluated the effect of a whey protein isolate coating incorporated with rice bran oil on the quality attributes of kiwifruit over 28 days of refrigerated storage. The results showed that the incorporation of rice bran oil into the coating assisted in reducing weight loss and maintaining sensory properties of the sample during storage. Uthairatanakij et al. [67] evaluated the impact of a whey protein isolate coating containing olive oil on the quality attributes of fresh-cut pineapple stored under refrigerated conditions for 8 days. It was found that the blend of whey protein isolate coating with olive oil was more effective in maintaining ascorbic acid and total phenolic contents of the sample than the pure whey protein isolate coating.

### 5.2. Meat Products

Meat and meat products are perishable food commodities that tend to deteriorate quickly under improper storage and transportation conditions. Edible films and coatings incorporated with functional additives provide great potential to minimize deteriorative processes in these products.

In this context, Zhou et al. [69] used a glucomannan/carrageenan coating incorporated with camellia oil to improve the quality of chicken meat throughout storage for 10 days under refrigerated conditions. Compared to the uncoated samples and the samples coated with a glucomannan/carrageenan-based coating, the samples coated with a glucomannan/carrageenan coating containing camellia oil displayed lower pH, weight loss, total volatile nitrogen, thiobarbituric acid reactive substances, and microbial counts during storage. Furthermore, it was found that the sensory attributes of chicken meat were not affected by the addition of camellia oil to the coating. Vargas et al. [71] applied a chitosan film containing sunflower oil to pork meat hamburgers and reported that the sample packed in the film containing sunflower oil had a significantly lower metmyoglobin content during storage (for 8 days under refrigerated conditions) than the sample packed in pure chitosan. Metmyoglobin, which is brown in color, is formed when myoglobin in meat is exposed to oxygen [72]. Wang et al. [68] studied the impact of a chitosan/potato protein film incorporated with linseed oil on the quality of pork meat stored for 7 days under refrigerated conditions. The authors found that adding linseed oil to the film helped to improve the quality of the pork meat during storage, as reflected by measuring its pH, total microbial counts, and sensory properties. In another study, the effect of hemp seed oil addition to a gelatin-based coating on the quality of pork meat over 12 of refrigerated storage was evaluated [70]. The authors in that study found that the pork meat coated with the gelatin coating containing hemp seed oil had better oxidative stability (lower thiobarbituric acid reactive substances and metmyoglobin content) and microbial quality (lower total aerobic plate count) than the uncoated samples and the samples coated with gelatin alone.

## 6. Challenges and Future Perspective

Incorporating POs in edible films and coatings can provide several advantages including enhanced barrier, antioxidant, and antimicrobial properties of the packaging material. However, there are a few challenges associated with their application. One major challenge of using POs as additives in edible films and coatings is their low solubility and limited compatibility with hydrophilic biopolymers. This limitation can result in uneven distribution of the PO within the packaging material and, thus, negatively influence the overall effectiveness of the film or coating. Furthermore, some plant oils, known for their distinctive flavors, may negatively influence the sensory properties of the coated food. To overcome these limitations, innovative approaches, such as nanoencapsulation, could be considered in future studies when incorporating POs in edible films and coatings. While POs have been shown to be effective in improving the performance of edible films and coatings, further studies are still needed to explore their stability within the packaging material and also their intermolecular interactions with the different components (such as biopolymers, plasticizers, and emulsifiers) of edible films and coatings. Finally, the majority of research incorporating Pos in edible films and coatings has been conducted at the laboratory scale. Therefore, in future studies, emphasis should be placed on scaling up these applications to assess their feasibility and effectiveness at the commercial level.

## 7. Conclusions

As a sustainable substitute to synthetic polymers, biopolymer-based packaging materials have gained significant attention in recent years. However, compared with conventional plastics, biopolymers alone have poor film-forming properties. In this context, blending biopolymers with Pos seems a promising strategy for improving the functionality of edible packaging materials. The incorporation of POs into edible packaging generally improves barrier capability (water vapor, oxygen, and UV light) and antioxidant and antimicrobial properties, while reducing the tensile strength of the packaging material. However, it is important to note that such effects could be highly influenced by several factors, including the composition and type of both the PO and biopolymer and the potential chemical interactions between the biopolymer and oil components. Therefore, selection of the PO and biopolymer with suitable formulations can significantly help in producing packaging materials with desired properties. Furthermore, according to previous studies, the addition of POs to edible packaging materials significantly helps in minimizing deteriorative processes in food products. However, migration of POs from films and coatings into food products should be further studied to ensure the sensory attributes of the packaged foods are not affected. Overall, POs show great potential as natural additives for improving the performance of edible packaging materials.

## Figures and Tables

**Figure 1 foods-13-00997-f001:**
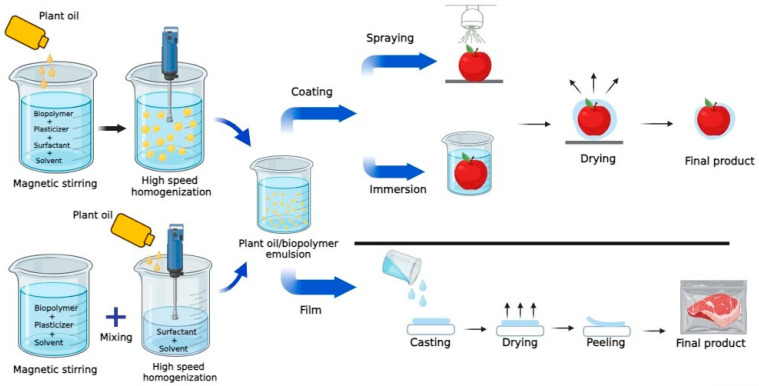
A simple representation of incorporating plant oils in edible films or coatings and their application in food packaging.

**Figure 2 foods-13-00997-f002:**
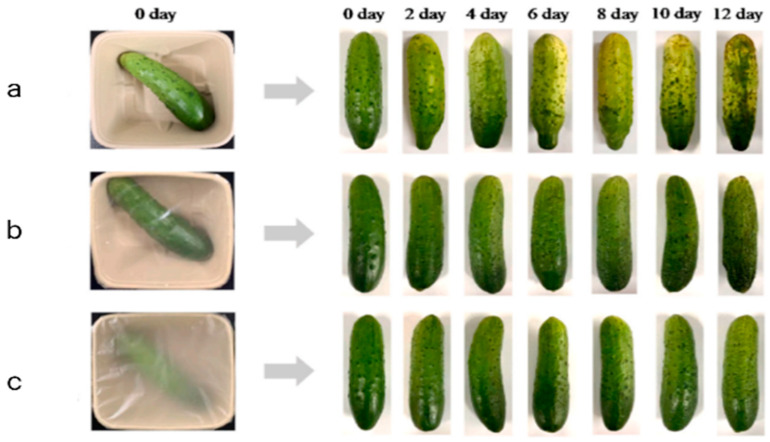
Visual appearance of the unpacked and packed cucumbers stored for 12 days at 7 °C: (**a**) unpacked cucumber; (**b**) cucumber packed using Konjak glucomannan/agar/gum Arabic film; (**c**) cucumber packed using Konjak glucomannan/agar/gum Arabic film + 0.4% coconut oil [16].

**Table 2 foods-13-00997-t002:** The influence of plant oil addition on the in vitro antimicrobial properties of edible films.

			Antimicrobial Activity		
Matrix	Plant Oils	Matrix: Oil Proportion	Tested Microorganisms	Results	Refs
Carboxymethyl cellulose	Black cumin seed oil	2% (*w*/*v*): 0–1% (*w*/*v*)	*Staphylococcus aureus*, *Escherichia coli*	The film containing black cumin seed oil showed higher antimicrobial activity against the tested microorganisms than the film without oil	[36]
Chitosan	Tea seed oil	1% (*w*/*v*): 0–0.5% (*w*/*v*)	*Botrytis cinera*	Oil addition significantly reduced the growth of *Botrytis cinerea*	[59]
Chitosan (medium MW)	*Berberis crataegina* seed oil	1% (*w*/*v*): 0–1.6% (*w*/*v*)	*Escherichia coli*, *Salmonella typhmurium*, *Proteus microbilis*, *Proteus vulgaris*, *Pseudomonas aeruginosa*, *Enterobacter aerogenes*, *Staphylococcus aureus*, *Streptococcus mutans*, and *Bacillus thuringiensis*	Antimicrobial activity of the film was improved against all tested microorganisms except *Proteus vulgaris*, *Enterobacter aerogenes*, and *Streptococcus mutans*	[37]
Chitosan (medium MW)	False flax seed oil	2% (*w*/*v*): 0–2% (*v*/*v*)	*Escherichia coli*, *Salmonella typhimurium*, *Proteus microbilis*, *Proteus vulgaris*, *Pseudomonas aeruginosa*, *Enterobacter aerogenes*, *Staphylococcus aureus*, *Streptococcus mutans*, and *Bacillus thuringiensis*	Oil incorporation improved the antimicrobial activity of the film against all the tested microorganisms	[33]
Chitosan (medium MW)	Olive, corn, or sunflower oil	1% (*w*/*v*): 0–0.25% (*v*/*v*)	*Escherichia coli*, *Staphylococcus aureus*, *Proteus microbilis*, *Proteus vulgaris*, *Pseudomonas aeruginosa*, *Enterobacter aerogenes*, *Bacillus thuringiensis*, *Salmonella typhimurium*, and *Streptococcus mutans*	Film with oil had higher antimicrobial activity than the film without oil The film incorporated with olive or sunflower oil showed higher antimicrobial activity than the film with corn oil	[54]
Pectin/gelatin	Grape seed or olive oil	5% (*w*/*v*): 0.75% (*w*/*w*)	*Salmonella typhimurium*, *Escherichia coli*, *Staphylococcus aureus*, and *Pseudomonas fluorescens*	The films containing grape seed or olive oil showed antimicrobial activity against all tested microorganisms	[20]
Gelatin	Camellia oil	3% (*w*/*v*): 0–100% (*w*/*w*, based on the mass of gelatin)	*Staphylococcus aureus* and *Escherichia coli*	The film with oil had higher antimicrobial activity against the tested microorganisms than the film without oil	[15]
Corn starch	Coffee oil	4% (*w*/*v*): 0–1% (*v*/*v*)	*Staphylococcus aureus*, *Escherichia coli*, and *Salmonella enterica*	The addition of coffee oil gave corn starch films antibacterial activity against the tested microorganisms	[14]
Potato starch	Coconut oil	2.5% (*w*/*w*): 0–112% (*w*/*w*, based on the mass of potato starch)	*Escherichia coli*, *Listeria monocytogenes*, and *Staphylococcus aureus*	The addition of coconut oil gave potato starch films antibacterial activity against the tested microorganisms	[25]
Gelatin	Corn oil	4% (*w*/*v*): 0–1.2% (*w*/*v*)	*Aspergillus niger*	Corn oil inhibited the antifungal activity of the film	[40]
Gelatin	Hemp seed oil	4% (*w*/*v*): 0–2% (*v*/*v*)	*Escherichia coli*, *Staphylococcus aureus*, *Listeria innocua*, *Saccharomyces cerevisiae*, and *Penicillium expansum*	The addition of hemp seed oil gave gelatin films antimicrobial activity against *Staphylococcus aureus* and *Listeria innocua*. Oil addition had no effect on the growth of *Escherichia coli*, *Saccharomyces cerevisiae*, and *Penicillium expansum*	[60]

**Table 3 foods-13-00997-t003:** The influence of plant oil addition to edible films and coatings on the quality parameters of different food products.

Matrix	Plant Oils	Matrix: Oil Proportion	Food Product and Packaging Conditions	Effect on Food Product	Refs
**Fruits and vegetables**					
Konjak glucomanan/agar/gum Arabic	Coconut oil	2.4% (*w*/*w*): 0–0.6% (*w*/*w*)	Cucumber, stored for 12 days at 7 °C; the biodegradable container containing the sample was covered with the film	Decreased weight loss and firmness reductions	[16]
Whey protein isolate	Rice bran oil	10% (*w*/*v*): 0–0.6% (*w*/*v*)	Kiwifruit, stored for 28 days at 4 °C, dipped in coating solution	Decreased weight loss; preserved firmness and taste; increased overall acceptability	[61]
Chitosan (MW: 96 kDa)	Tea seed oil	1% (*w*/*v*): 0–0.1% (*w*/*w*)	Strawberry, stored for 24 days at 2 °C, dipped in coating solution	Reduced weight loss; retained firmness, color, moisture content and total soluble solids; delayed pH changes	[62]
Wild sage gum	Pomegranate seed oil	0.1–0.2% (*w*/*v*): 0–0.05% (*w*/*v*)	Mexican lime fruit, stored for 24 days at 20 °C, dipped in coating solution	Decreased weight loss; preserved total phenols, flavonoids, color, antioxidant capacity, and sensory properties	[63]
konjac glucomannan/curdlan	Camellia oil	1% (*w*/*v*): 0–0.15% (*w*/*v*)	‘Kyoho’ grapes, stored for 10 days at room temperature, dipped in coating solution	Maintained the appearance, total soluble solids, and acid content; delayed weight loss and firmness decrease	[64]
Carboxymethyl cellulose	Pomegranate seed oil	3% (*w*/*v*): 0–3% (*v*/*v*)	Strawberry, stored for 16 days at 5 °C, dipped in coating solution	Decreased weight loss; maintained highest level of total phenolic content	[65]
Chitosan	Tea seed oil	1% (*w*/*v*): 0–0.5% (*w*/*v*)	Pear fruit, stored for 21 days at 25 °C, dipped in coating solution	Reduced respiration rate and fungal decay; maintained total soluble solids	[59]
Soy protein isolate	Olive oil	2–6% (*w*/*v*): 0.7–1.1% (*v*/*v*)	Pear fruit, stored for 5 days at 28 °C, dipped in coating solution	The weight loss of the sample decreased as the olive oil concentration in the coating increased	[66]
Whey protein isolate	Olive oil	10% (*w*/*v*): 0–1% (*v*/*w*)	Fresh cut pineapples, stored for 8 days at 4 °C, dipped in coating solution	Maintained ascorbic acid and total phenolic contents	[67]
**Meat products**					
Chitosan/potato protein	Linseed oil	2.6%: 0–33%	Pork meat, stored for 7 days at 4 °C, wrapped with the film	Decreased change in pH; reduced microbial growth; preserved sensory properties	[68]
Konjak glucomannan/carrageenan	Camellia oil	1% (*w*/*v*): 0–3.5% (*w*/*v*)	Chicken meat, stored for 10 days at 4 °C, dipped in coating solution	Decreased change in pH; reduced weight loss, total volatile nitrogen, thiobarbituric acid reactive substances, and microbial growth Extended shelf life of the chicken meat to 10 days by retarding the oxidation of lipids and proteins and microbial growth	[69]
Gelatin	Hemp seed oil	4% (*w*/*v*): 0–2% (*w*/*v*)	Pork meat, stored for 12 days at 2 °C, dipped in coating solution	Improved oxidative stability; reduced microbial growth	[70]
Chitosan (high MW)	Sunflower oil	1% (*w*/*w*): 0–1% (*w*/*w*)	Pork meat hamburger, stored for 8 days at 4 °C, coated with the film	Decreased formation of metmyoglobin	[71]

## Data Availability

The original contributions presented in the study are included in the article, further inquiries can be directed to the corresponding author.

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
