# Peer review of "Application of Plant Oils as Functional Additives in Edible Films and Coatings for Food Packaging: A Review"

_foods, 2024, doi:10.3390/foods13070997_

Round 1
Reviewer 1 Report
Comments and Suggestions for Authors
Minor improvement suggestions following:
Lines 24-25: reducing the growing food waste generation
Line 33:” utilization of them” should be rephrased
Line s55-56: “essential oils and plant oils (POs)” Considering that essential oils are ussually coming from from plants an explanation of the difference between the two oil categories is mandatory.
Lines 63-64: “POs have numerous bioactive compounds in their composition which are responsible for their antioxidant and antimicrobial activities” As the bioactive compounds are usually present in essential oils and not in POs, some examples would clarify this aspect
Lines 162-163 and elsewhere in the text: concentration reported to w/w or vol should be indicated
Acknowledgments: This is contribution no. XX-xXX-J from the Kansas Agricultural Experiment 431 Station, Manhattan, KS.
Is it correct the contribution no?
Reviewer 2 Report
Comments and Suggestions for Authors
Overall, the review are more like a lists of literatures in the forms of table and introduction the results of various studies. I can’t see the thoughts of the authors in the review. They used the pattern that PO has effect on one property, some exhibited the effect in good, some exhibited the effect is bad. I can’t regard the review as a qualified academic review. The authors should think more what points can be summarized for the literatures, and what points you want to convey to readers.
Figure 1: the author presented several methods for food films/coatings production, which is good. However, I suggest the author make it more detailed based on the reference. Some studies prepared emulsions first and then made food films or coatings or other methods has been applied before film making. Can you include the process on the schematic illustration?
Section 3.1: the effect of plant oil on the thickness of the films is not essential in a discussion of review because the thickness of the films is not only up to the composition but also the handling process and handling people. So the discussion here is meaningless.
Table 1: the abbreviations should be explained.
Section 3.9 and 3.10: the antioxidant and antimicrobial properties are very important properties for active package making. The authors should more focus on discussing the effective molecules in plant oils that devote antioxidant or antimicrobial activities.
The authors should add a section to discuss the challenges and prospects of the plant oil encapsulated food package.
Reviewer 3 Report
Comments and Suggestions for Authors
The document titled “Application of plant oils as functional additives in edible films and coatings for food packaging: A review” presents a very clear review of the advantages of using these packaging or coatings and its content is of scientific interest. However, I have listed some suggestions that could improve the document substantially.
-It would be good to add a section on the interactions of the oil with the biopolymer
-Include the minimum mechanical properties necessary for the wrappers and, in the case of coatings, their minimum stability (environmental factors, air oxidation, light, etc.)
-Molecular weight of biopolymers, it would be good to mention them
-How long do plant oils added to the formulation maintain their properties and are they capable of changing the organoleptic properties of foods?
-What techniques or equipment are used to generate these packaging or coatings?
Round 2
Reviewer 2 Report
Comments and Suggestions for Authors
The revised version is acceptable.